# Plasma Proteomic Variables Related to COVID-19 Severity: An Untargeted nLC-MS/MS Investigation

**DOI:** 10.3390/ijms24043570

**Published:** 2023-02-10

**Authors:** Lisa Pagani, Clizia Chinello, Giulia Risca, Giulia Capitoli, Lucrezia Criscuolo, Andrea Lombardi, Riccardo Ungaro, Davide Mangioni, Isabella Piga, Antonio Muscatello, Francesco Blasi, Andrea Favalli, Martina Martinovic, Andrea Gori, Alessandra Bandera, Renata Grifantini, Fulvio Magni

**Affiliations:** 1Proteomics and Metabolomics Unit, School of Medicine and Surgery, University of Milano-Bicocca, 20854 Vedano al Lambro, Italy; 2Bicocca Bioinformatics Biostatistics and Bioimaging Centre—B4, School of Medicine and Surgery, University of Milano-Bicocca, 20854 Vedano al Lambro, Italy; 3Department of Pathophysiology and Transplantation, University of Milano, 20122 Milano, Italy; 4Infectious Diseases Unit, IRCCS Ca’ Granda Ospedale Maggiore Policlinico Foundation, 20122 Milano, Italy; 5Respiratory Unit and Cystic Fibrosis Adult Center, Internal Medicine Department, Foundation IRCCS Ca’ Granda Ospedale Maggiore Policlinico, 20122 Milano, Italy; 6Istituto Nazionale di Genetica Molecolare (INGM), 20122 Milano, Italy

**Keywords:** COVID-19, proteomics, mass spectrometry, SARS-CoV-2, plasma, blood, severe

## Abstract

Severe Acute Respiratory Syndrome Coronavirus-2 (SARS-CoV-2) infection leads to a wide range of clinical manifestations and determines the need for personalized and precision medicine. To better understand the biological determinants of this heterogeneity, we explored the plasma proteome of 43 COVID-19 patients with different outcomes by an untargeted liquid chromatography-mass spectrometry approach. The comparison between asymptomatic or pauci-symptomatic subjects (MILDs), and hospitalised patients in need of oxygen support therapy (SEVEREs) highlighted 29 proteins emerged as differentially expressed: 12 overexpressed in MILDs and 17 in SEVEREs. Moreover, a supervised analysis based on a decision-tree recognised three proteins (Fetuin-A, Ig lambda-2chain-C-region, Vitronectin) that are able to robustly discriminate between the two classes independently from the infection stage. In silico functional annotation of the 29 deregulated proteins pinpointed several functions possibly related to the severity; no pathway was associated exclusively to MILDs, while several only to SEVEREs, and some associated to both MILDs and SEVEREs; SARS-CoV-2 signalling pathway was significantly enriched by proteins up-expressed in SEVEREs (*SAA1/2*, *CRP*, *HP*, *LRG1*) and in MILDs (*GSN*, *HRG*). In conclusion, our analysis could provide key information for ‘proteomically’ defining possible upstream mechanisms and mediators triggering or limiting the domino effect of the immune-related response and characterizing severe exacerbations.

## 1. Introduction

Since December 2019, Coronavirus disease 2019 (COVID-19) has been spreading around the world. Subjects infected by severe acute respiratory syndrome coronavirus-2 (SARS-CoV-2), the etiologic agent responsible for COVID-19, can develop a wide range of symptoms. Indeed, patients can range from an asymptomatic state to showing life-threatening clinical signs, such as the development of acute respiratory distress syndrome (ARDS). In more detail, it has been estimated that 20% of patients develop respiratory problems that require oxygen supply; some of them also suffer from neurological or haematological problems, while the mortality rate has been calculated as 2.3%. It is known that elderly people and subjects with comorbidities are at more risk to develop severe symptoms, and that the disease course could be influenced and dependent on several predisposing factors (such as diet, environment, genetic background) [1,2,3]; however, the molecular mechanisms responsible for the worsening of the conditions are still not fully understood [4,5,6]. 

In this context, blood proteomics shows a great potentiality, since this biofluid can be collected in a less invasive way and reflects the changes of the whole organism [7,8,9]. It has been shown by supervised learning that the plasma proteome is a highly informative indicator of the clinical severity of COVID-19 patients, superior to other sources and techniques such as the transcriptome of the peripheral blood mononuclear cells (PBMCs) [10]. Therefore, in these two years of pandemic, blood collected from patients infected with SARS-CoV-2 has been widely studied with mass spectrometer (MS)-based proteomics techniques in order to investigate COVID-19 disease more deeply. It has indeed been reported that the plasma proteome of patients affected by COVID-19 remains perturbed for at least 6 weeks after the first positive swab test. Furthermore, by the analysis of plasma proteins at the time of seroconversion, it can be possible to detect which patients will suffer from post-acute sequelae of SARS-CoV-2 (PASC) [11].

In the literature, some proteins have been identified as the most commonly deregulated in COVID-19 patients. They include: *SAA2*, *SAA1*, *ITIH3*, *LBP*, *LGALS3BP*, *CFB*, *CRP* and *APOA1*. Among them, *CRP* is usually known as a marker of inflammation, while, in a recent work, the expression of the first four proteins on the list have been observed to increase in correlation to the severity of the disease. Regarding *LGALS3BP*, *CFB* and *ITIH4*, they are more expressed in all COVID-19 patients as compared to healthy subjects; on the other hand, *APOA1* shows the opposite pattern, a decreasing expression in uninfected individuals [12]. 

Since the first works were published, a correlation between blood proteome and disease severity was observed. Park et al. analysed a small cohort of patients and they found a deregulation of neutrophil activation and blood coagulation pathways by comparing a group of mild patients with severe ones [13]. A similar study was performed by Messner et al. with serum and plasma samples collected from a larger cohort; they identified a list of 27 possible biomarkers associated with COVID-19 severity. These proteins are involved in different pathways, which include: complement factors, coagulation system, inflammatory modulators and proinflammatory signalling [8].

Moving to more recent works, Ciccosanti et al. found a panel of proteins that are mainly involved in the acute inflammatory response and that are increased in all COVID-19 patients, but more expressed in patients who required admission to an intensive care unit (ICU) or who had a fatal outcome. Instead, other proteins are upregulated only in patients with a more severe outcome, they are mainly involved in the complement cascade, in the coagulation pathway, in the extracellular matrix organisation and they also include proteins involved in some types of amyloid diseases and VLDL/LDL lipoproteins [4]. However, the role of the complement and coagulation cascade in the disease still remains ambiguous, since Captur et al. observed an increase in this pathway in patients with non-severe infection, in addition to lipid atherosclerosis and cholesterol metabolism, lysosomal function and autophagy pathways. They also reported that the proteins involved in COVID-19 disease during the acute infection mainly include markers of oxidative stress, metabolic reprogramming factors and cell adhesion molecules [11].

Recognizing the molecular variables associated with worse outcomes could be pivotal for obtaining the knowledge required to plan an effective triaging and reshape the personalised care of each patient, allowing a better administration of human and material resources [14]. The discovery of the molecular signature of the disease in relation to its progression and the intensity of clinical manifestation would also enable the identification of possible therapeutic targets [15]. Variations in human plasma protein abundances, thus, have been so far commonly recognized as effective indicators of pathophysiological states and changes in a plethora of diseases, including various virosis [8].

In this scenario, our goal is to investigate by liquid chromatography tandem mass spectrometry (LC-MS/MS) the plasma proteome of COVID-19 patients with different outcomes, from less to more severe ones. Thereby, we aim to define which key protein patterns and biological networks result differently modulated in response to the interaction of SARS-CoV-2 with the host and explore the chain mechanistic effects stimulated by the deregulation of these altered proteins that leads to a gap in the phenotype spectra.

## 2. Results

An untargeted and label-free proteomic approach has been applied to plasma samples of patients affected by SARS-CoV-2 presenting different outcomes. Based on the WHO COVID-19 Severity Index and clinical criteria, a selected group of patients (n = 43) were divided into two subgroups (MILDs and SEVEREs) for the following statistical analysis. The Mild group basically includes asymptomatic or paucisymptomatic ambulatory treated subjects while the Severe class regards hospitalised patients in need of oxygen support therapy. The WHO classification was referred to the day of sample collection. A description of the characteristics of the enrolled patients is reported in Table 1. The p-value referred to the non-parametric test on the Age variable is significant. While no significant difference based on gender between Severe and Mild patients was highlighted by Fisher’s exact test. To exclude a possible hidden confounding interference of these two variables, all the statistical analysis and comparisons are additionally adjusted for these factors, as mentioned in the Materials and Methods section.

### 2.1. Proteomic Variables Related to Worse Outcome

Aimed at the study of plasma proteins whose expression is associated with the clinical manifestations of disease conditions triggered by SARS-CoV-2, we proteomically explored plasma samples collected from COVID-19 patients with different outcomes. By a label-free nLC-MS/MS based approach, 554 human proteins were identified in the two groups of patients (MILDs & SEVEREs); among them, a subset of 331 proteins that met more stringent quality criteria were also considered for a relative quantification. All of the protein IDs are listed and detailed in Appendix A.

As a first step, multiple statistical comparisons were applied to investigate the proteome changes associated with the severity of the disease. A volcano plot of the significance related to Mild and Severe patients was obtained (Figure 1). The graphic highlights the dysregulation of a panel of 29 proteins whose expression was strongly influenced by the intensity of the symptoms (*p*-value ≤ 0.05; Fold change ≤ −1.5 or ≥ 1.5). In particular, 17 of them were upregulated in patients manifesting worse outcomes. The complete list of these differentially expressed proteins (DEPs) is illustrated in Appendix A.

All of the 29 proteins belonging to this panel were used to perform a heatmap-cluster analysis (Figure 2) in order to evaluate the similarities between the protein profiles for each patient. In the graphic, the different abundances of each protein in correlation with each single patient are visually shown. From the heatmap, a double and mutual connection can be observed that confirms and supports the specificity of the molecular signatures underlined by the multiple comparison analysis: (i) an evident clusterization of the MILDs from the SEVEREs on one side, and (ii) a clear clusterization of all the DEPs consistent with their expression in the two groups of patients on the other side. Only two patients out of forty-three are likely to mis-clusterize, a behaviour probably ascribed to possible confounding factors, including biological variability. 

### 2.2. Classification Tree Related to Severity

The presence of significant molecular signatures associated with the severity suggested the idea of checking the most significant proteins able to characterise the different levels of infection in the patient cohort. To this aim, a supervised analysis based on a classification tree was applied [16].

The classification tree in Figure 3 shows that the first protein that splits the patients into two groups is FETUA (*AHSG*—Alpha-2-HS-glycoprotein), with a cut-off of 25 × 10^6^ In the right arm, corresponding to the patients with the FETUA area < 25 × 10^6^ (16% of the samples), the tree classifies as SEVEREs 70% of the Severe patients. Instead, in the left arm, it recognizes all the Mild patients and three Severe patients as MILDs (84% of accuracy). In a second step, the condition VTNC (*VTN*—Vitronectin) ≥ 26 × 10^6^ increases the accuracy of classification to 98%. In the third and last split, the node LAC2 (*IGLC2*—Ig lambda-2 chain C regions) ≥ 5.7 × 10^6^ identifies the last two leaves of the tree, where all the Severe patients are clustered as SEVEREs and, on the other hand, all the Mild ones as MILDs. 

The same analysis was repeated considering a smaller cohort of patients, which included only the patients whose plasma was collected in the acute phase within 21 days from the first positive test (patients with negative test excluded). The description of this subgroup of patients is reported in Appendix A. In the classification illustrated in Figure 4, the highest accuracy was reached in only two steps based on FETUA and LAC2 proteins, additionally confirming the role of these two molecules in the onset and in the first stages of the disease development.

### 2.3. Functional Annotation

Bioinformatic pipelines were applied for in silico functional annotation in order to recognise a panel of proteomic functional signatures and uncover possible altered biological processes and pathways that differently characterise COVID-19 patients with mild or severe outcomes.

The panel of 29 DEPs pinpointed in the multiple comparisons analysis was used to perform the functional enrichment and highlight possible functions associated with severity.

The biological processes resulting from upregulated proteins in severe or mild patients are listed in Appendix A, respectively. In general, a greater involvement of both innate and adaptive immune responses, including the classical complement activation and the leukocytes- and neutrophils-mediated immunity, was observed in patients that were hospitalised, who needed oxygen supplement as compared to asymptomatic or paucisymptomatic patients (Appendix A). On the other hand, this last group was likely to manifest a more evident enhancement of the platelet activation and degranulation processes (Appendix A). 

Reactome pathway analysis was carried out on the entire panel of 29 DEPs and schematised in Appendix A. Interestingly, no pathway is associated only with MILDs, while several pathways are associated only to the SEVEREs group or shared by both MILDs and SEVEREs. In particular, only pathways related to haemostasis, the complement cascade, and partially to vesicle mediated transport, seem to be altered in patients with no or mild symptoms as well (Figure 5). On the other side, the enrollment of a plethora of defence responses, including the pathways typical of the adaptive immunity (as the signalling of B cell receptor) and others, specific of the humoral innate responses (i.e. Fc epsilon receptor (FCERI) signalling and Fc gamma receptor (FCGR) dependent phagocytosis) were triggered in hospitalised patients in need of oxygen. It should be noted that the involvement of the complement cascade differs from MILDs and SEVEREs, for which the enhancement of the protein level is likely to be related not only to the regulation of this pathway, but also to the triggering of the classical antibody-mediated complement pathway (Figure 5). Regarding the deregulation of the vesicle mediated transport, except for the involvement of *RAB1A* up-expressed in MILDs, the scavenging of heme from plasma seems to be enforced in patients with a severe clinical manifestation (Appendix A).

Using a different data source for pathway enrichment, in addition to the already observed involvement of the complement cascade both in SEVEREs and MILDs, a deregulation of a specific network map of SARS-CoV-2 signalling pathway for both key proteins whose concentration has been shown to increase both in patients with severe and mild symptoms was highlighted *(SAA1/2*, *CRP*, *HP*, *LRG1* and *GSN*, *HRG*, respectively) (Appendix A). Through this annotation approach, a stimulation of proteins related to Vitamin B12 and folate metabolisms and selenium micronutrient network was also noted. 

## 3. Discussion

### 3.1. Key proteins Characterising Patients Based on the Intensity of the Symptoms

Following the outbreak of COVID-19 and its declaration as a pandemic on 11 March 2020, much work has been performed in order to unravel the mechanisms which underpin this disease. In particular, proteomics and related technologies have been readily used due to their capacity to underline molecular alterations relevant to its pathogenesis directly from easily obtained biological specimens, such as blood. These more in-depth molecular insights into the upstream molecular processes responsible for the downstream clinical display of COVID-19 could shed light on the molecular mechanisms deriving from the immune-related response to this pathogen.

Herein, a panel of 29 DEPs was identified, of which, 17 were over-expressed in severe patients and 12 in MILDs (Figure 1; Appendix A). Among them, several proteins showed an expression trend that appears in line with the literature. Ciccosanti et al. observed an increase of the serum amilod A1 and A2 (SAA1, SAA2) proteins in patients who require admission to an ICU (intensive care unit), compared to patients who do not require ICU admission and to healthy subjects [4]. This observation is consistent with the significant increase of their abundance in the plasma of patients with worse outcomes. Their correlation with disease severity is also likely to be confirmed by the work of Sahin et al. [12]. *CRP* and *SERPINA3*, which are considerably more concentrated in the plasma of SEVEREs, have already been pointed out as candidate predictors for worse outcomes in COVID-19 [4].

The application of a DT strategy allowed for the definition of three crucial proteins that were able to distinguish SEVEREs and MILDs for all of the subjects belonging to the cohort.

Fetuin-A (α-2-Heremans-Schmid glycoprotein, FETUA-*AHSG*), independent of the infection phase in which the plasma was collected, has been shown to represent a key molecule for classifying patients based on the outcome. In particular, on its own, it has been able to efficiently separate the two groups of patients based on severity, obtaining 97% or 94% accuracy, depending on the step of disease progression (Figure 3 and Figure 4). Its plasma level has been observed as significantly halved in severe patients (Appendix A). This negative regulation of *AHSG* in COVID-19 patients and its possible substantial role in the processes leading to the exacerbation of SARS-CoV-2 infection is robustly confirmed in the literature. Fetuin-A is a 60 KDa glycoprotein with a normal serum value of 300–600 g/L and a pleiotropic role. In particular, its involvement in immune response regulation and inflammation has been mostly ascribed to its behaviour as a liver-derived negative acute phase reactant, so far reported [18]. It is known that Fetuin-A serum concentration indeed decreases during the acute inflammatory response via inhibition of cytokines such as interleukin-6 (IL-6) and tumor necrosis factor-alpha (TNF-α), and it returns to normal values after successful treatment in many systemic and non-inflammatory diseases, such as axial spondyloarthritis and inflammatory bowel disease [19], rheumatoid arthritis [20], inflammatory CNS disease [21] and also in other respiratory pathologies, i.e., chronic obstructive pulmonary disease (COPD), for which it has been considered a candidate biomarker for the prediction and evaluation of worsening conditions [22]. Kukla et al. showed for the first time a significantly lower level of this hepatokine in the serum of COVID-19 patients and advanced a possible role of its deficiency in the development of cytokines storm during SARS-CoV-2 infection and in predisposing to a more severe disease course, independently from confounding factors such as sex, metabolic disorders, lipids level, BMI, respiratory symptoms, or liver injury [23]. They also observed a significant decrease in the abundance of this protein in COVID-19 patients with pneumonia and in those who required critical care, indicating that lower fetuin-A levels could influence the disease course and may be associated with a predisposition for worse prognosis. More recently, Reverté L. et al. determined *AHSG* together with inter-α-trypsin inhibitor 3 (*ITIH3*) as the most accurate biomarkers of the critical clinical progression of COVID-19 by random forest modelling [24]. Herein, the importance of Fetuin-A in distinguishing mild patients from more critical ones is confirmed in an Italian Caucasian population independently from the infection state.

The strength of this classification operated through the DT is further enforced by the use of two other proteins which are both more abundant in SEVERE’s plasma than in MILD’s: vitronectin and Immunoglobulin lambda constant 2, which annul any possible mismatch or error in group recognition (Figure 3 and Figure 4). 

Vitronectin (VTNC-*VTN*), in particular, a 75 KDa multifunctional glycoprotein also termed the S-protein of the complement system, is produced predominantly by the liver and has a serum concentration of 200–400 μg/mL. It plays a crucial role both in tissue remodeling by regulating cell adhesion through binding of its Arg-Gly-Asp (RGD) motif to different types of integrins, and also in the regulation of the blood system related protease cascades, such as coagulation and fibrinolysis via heparin and thrombin-antithrombin III complexes [25]. A recent study showed that *VTN*, together with other proteins involved in the extracellular matrix organization, were highlighted as belonging to a cluster of molecules that are significantly upregulated only in COVID-19 patients with fatal pneumonia compared to those with severe pneumonia requiring ICU admission, and to subjects with pneumonia that do not require ICU admission [4]. Interestingly, this protein has been also reported to be recruited together with Clusterin by coronavirus-infected cells from human serum through Antibody-Dependent Mechanisms, and to be associated with delayed Complement-mediated death, provoking a series of implications related to viral pathogenesis and tissue tropism [26].

On the other hand, the antibody IGLC2 (LAC2-*IGLC2*) unlike vitronectin seems to confirm its impact in severity discrimination also in acute stages of the disease, and shows an expression behaviour in line with the literature. One of the most observed COVID-19-specific signatures at the transcriptional level concerns immunoglobulin-related genes [27], including IGLC2, which were found to be consistently upregulated in correlation to COVID-19 disease across different datasets [28,29]. Makund et al. demonstrated that a significant common program of transcriptional dysregulation of immunoglobulin genes, including *IGLC2* and *IGHA1/IGHM/IGKC/IGLC3*, exists across both the myeloid and lymphoid milieu of COVID-19 patients, even if they do not differ across severities [30]. Moreover, IGLC2 has been highlighted as upregulated in the form of transcript in B-cells of SARS-CoV-2 patients [29], and as a protein in sera of COVID-19 patients, where its level has not been normalised after patient recovery [31].

### 3.2. Functional Signatures of Severity

One of the critical challenges related to SARS-CoV-2 infection still remains the uncovering of mechanistic pathways encompassing the immunological imbalance and clinical complications arising from the severe forms of this retrovirus.

Specific immune signatures related to severe host responses appeared evident also by the functional pathway enrichment. Hallmarks of disease severity seem to be mainly correlated to the inflammatory mediators and networks, including acute phase response proteins, leukocyte mediated immunity and neutrophil degranulation (Appendix A). Several proteins linked to the acute phase response, such as *CRP*, *SERPINA3*, *SAA1* and *SAA2*, have already been shown as increased in sera of severe COVID-19 patients, through an MS based Data Independent Analysis by Lee et al. [32]. *ORM1* and *HP* were also highlighted as altered depending on the severity of the patients by Shen et al. in serum [33] and by Beimdiek J et al. in plasma, respectively [34]. The modulation and the polymorphism of the human leukocyte antigen (HLA) play a key role in the immune response, and its variants could affect COVID-19 progression and severity [35,36]. Additionally, a gradual augment depending on the disease severity of the neutrophil degranulation has been revealed by a trans-omics study performed by Wu et al. [37].

The over-inflammatory state detected in COVID-19 is known to deal with numerous players that are recruited and mobilized, especially during exaggerated immune responses and worse outcomes [38]. This immune hyper-activation could thus be considered a key driver of COVID-19, even if the mechanisms that lead to it still remain uncertain. In particular, the severe phenotypes of COVID-19 carried the triggering of multiple interdependent events related to complement activation, dysregulated neutrophilia, endothelial injury and hypercoagulability [39], consistent with what we observed (Appendix A). In addition it was demonstrated that the response to SARS-CoV-2 could be a result of the imbalance of controlling virus replication versus the activation of the adaptive immune response, and also that a diminished innate antiviral defence, combined with a boosted production of the inflammatory cytokines, could define the clinical profile of COVID-19 [40]. An involvement of the adaptive defence, particularly of the B cell receptor (BCR) signalling, and the related up-regulation of several immunoglobulins has been highlighted (Appendix A and Figure 5). How the adaptive immune responses between severe and mild SARS-CoV-2 patients are differentially modulated and the role of B cells in the progression of this disease remain largely unknown; however, an effect of SARS-CoV-2 infection on BCR signalling has been demonstrated, operating through the alteration of the metabolomic and transcriptome profiles of B cells [41]. Moreover, increased levels of BCR clonal expansion and B-cell activation have been observed in patients displaying serious outcomes, indicating a more robust humoral immune response associated with the severity of symptoms [42]. Higher levels of a selected panel of immunoglobulins in SEVEREs, as highlighted in Appendix A, likely suggests possible implications of the innate immune response via activation and signalling of Fc gamma and Fc epsilon receptors (FCGR and FCERI). It has been recently reported that the difference in SARS-CoV-2-specific antibodies’ ability to elicit Fc-mediated innate immune functions could represent a candidate contributor for outcome exacerbations and an inflammation state, supporting the observed hyper modulation of these pathways in patients with worse symptoms [43]. Proteins related to scavenging of heme in plasma were also found in our data as upregulated in SEVEREs (Figure 5) (*HP*, *IGHV1-46*, *JCHAIN*, *IGKV3-20*, *IGKV3-11*, *IGLC2*, *IGLV3-1*, *IGLV3-21*). The levels of hemolytic products, such as free heme, were shown to induce neutrophil extracellular trap (NET) and damage-associated molecular patterns associated with the severity of septic patients [44], making hemolysis a marker of disease progression and a therapeutic target [45].

### 3.3. Functional Patterns Associated to Both SEVEREs and MILDs

The enrichment of pathways for DEPs particularly related to immunomodulation was expected and is consistent with the literature [6]. However, it is curious to notice that no systemic processes enriched with functional annotation analysis appeared related only with molecules with higher levels in MILDs. This remarkable functional behaviour is partially supported by several findings related to this disease.

The dysregulation of the hemostasis process in our analysis emerged as driven by some proteins that are more abundant in SEVEREs (*IGHV1-46*, *ORM1*, *SERPINA3*, *IGLC2*, *IGKV3-20*, *IGLV3-19*, *JCHAIN*, *IGKV3-11*, *IGLV3-21*) and others over-concentrated in MILDs (*ITGB3*, *HRG*, *FLNA*, *AHSG*, *GP9*) (Appendix A). SARS-CoV-2 has been shown to affect the disruption of the coagulation system, including the excessive activation of the platelets, leading to hypercoagulation and thrombotic events [46]. In particular, in COVID-19 patients manifesting severe symptoms, a loss of immune homeostasis has been observed and ascribed to a deficient immune response or, on the contrary, to its overstimulation [47]. Thus, it is reasonable to assume that the different modulation of critical nodes in this cascade and in its regulation can be displayed in consequence to different phenotypes. 

The important role of complement (C’) pathways in COVID-19 is well known and studied. Its involvement was confirmed from our investigation and can be seen both in MILDs and SEVEREs. In particular, the alteration of the abundances of proteins related to the classical antibody-mediated path through the creation of C2 and C4 was shown (Appendix A), and appeared in line with the COVID-19 literature [48]. The regulation of C’, on the other hand, encompasses *CFHR4/ITGB3* upregulated in MILDs, suggesting a potential role of these molecules in the successful control of COVID-19 infection (Appendix A). Of note, a case report demonstrated that the pathogenic nature of IgG4 autoantibodies directed against CFH could trigger the C’-mediated thrombotic microangiopathy in a patient carrying a genetic predisposition for homozygote *CFHR1/4* gene deletion [49]. On the other hand, *ITGB3*, together with *ITGA2B*, were shown to be potential intervention targets for COVID-19 stroke, due to their direct involvements in processes related to this outcome, as integrin signalling, and the response to elevated platelet cytosolic Ca^2+^, the consequent regulating platelet activation, the extracellular matrix- (ECM-) receptor interaction, the PI3K-Akt signalling pathway, and the hematopoietic cell lineage [50].

### 3.4. Specific Enrichment in SARS-CoV-2 Signalling Pathway

The picture outlined above, related to the modulation of the pathways based on the severity, seems to also be corroborated by other observations related to the variation of the abundances of proteins more specifically connected to downstream molecular signalling events triggered by the interaction host/SARS-CoV-2 (Appendix A), a pathway that resulted strongly enriched in our analysis (Appendix A). Remarkably, protein isoforms upregulated in SEVEREs (as *SAA1/2*, *CRP*, *HP*, *LRG1*) have indeed been shown to map only in virus-mediated pathways with a positive regulation of gene/protein expression. On the contrary, gelsolin (*GSN*) and histidine-rich glycoprotein (*HRG*) associated with an increased plasma level in MILDs were found in pathways that are substantially inhibited in SARS-CoV-2 host response expected to be negatively regulated (Appendix A).

This behaviour apparently suggests that when the clinical manifestation is moderate, a strengthening of negative modulators could occur and plays a critical role, potentially protective of the disease exacerbations. Gelsolin has been known to be an active scavenger, able, when upregulated, and together with DNase I, to compensate for the actin-mediated inhibition and influence the neutrophil extracellular traps (NET) clearance whose depowering was associated with severe COVID-19 pneumonia and higher mortality [51]. Plasma gelsolin (pGSN) has been shown as downregulated in individuals with active COVID-19 disease as compared to healthy subjects [8,52], and very recently, it has been demonstrated that it can be used in a combination with multiple analytes as a significant predictor of COVID-19 hospitalisation and poor outcomes [53]. Of note, the deficiency of plasma *GSN* could significantly impair its known organ-protective function, causing complications frequently present in individuals affected by COVID-19, such as multi-organ dysfunction syndrome (MODS), to higher levels of mortality and to long-term morbidity in survivors [54]. Interestingly, HRG shows a similar behaviour in relation to inflammatory states of disease. Low HRG levels have been associated with COVID-19 patients, and its decrease could lead, just like pGSN, to multiple organ failure through a chain of events that include the demodulation of the coagulation–fibrinolysis system, an abnormality of neutrophil morphology and endothelial cells, and a consequent immune thrombosis [55,56]. Moreover, its decrease could be a possible predictor of the mortality risk in severe COVID-19 cases, being consistently more abundant in survivors than in non-survivors [57].

### 3.5. Limitations

Our study presents some limitations. The major limitation is due to the timing of the collection of samples, which occurred at the beginning of the pandemic. Not all of the metadata of all of the patients was collected at that time, which prevented us from performing a correlation between our analysis and possible co-morbidities or treatments, and from evaluating other inferences. 

However, our findings found strong evidence in the literature, as detailed in the Discussion section, both at the molecular and functional level, and the partial heterogeneity of our cohort supported our aim to focus more on exploring specific and holistic protein signatures of the severity than on studying the disease progression. Moreover, the ability of few key proteins to recognise the cohort based on the disease phenotype was maintained independently from the collection time, and from confounding interferences such as age and gender, indirectly enforcing the involvement of these molecules in driving and/or measuring the levels of symptom exacerbation. 

## 4. Materials and Methods

An overview of the experimental design is illustrated in Figure 6.

### 4.1. Plasma Sample Collection

Plasma samples from COVID-19 patients were collected at the Foundation IRCCS Ospedale Maggiore Policlinico, Milano, Italy, from patients hospitalised for COVID-19 or evaluated in the outpatient clinic. Blood samples were collected in Vacutainer^®^ K3E tubes containing EDTA (Becton Dickinson Italia S.p.A., Milano, Italy), centrifuged at 3700 rpm for 10 min and then stocked at −80 °C. The study was approved by the Institutional Review Board Milano Area 2 (#103388-1 April 2020). All participants provided their written informed consent to participate in this study, which was conducted in accordance with the Declaration of Helsinki.

From the entire cohort, 43 patients met specific clinical criteria, were selected and subjected to data analysis and, in case of multiple sample collection, only one sample was considered. The 43 patients were divided into two groups: “Mild” and “Severe” on the basis of the gravity score assigned by the World Health Organization (WHO), according to Appendix A [8]. The gravity score was referred to the time of the collection. In particular, ambulatory asymptomatic or paucisymptomatic patients with no limitation of activity were considered as MILDs, while hospitalised patients in oxygen therapy without intubation or mechanical ventilation were considered as SEVEREs. A clinical description of the population is reported in Appendix A.

### 4.2. Sample Inactivation and Deglycosylation

To allow safe sample manipulation, the virus was inactivated and plasma samples were digested by following the protocol described by Pagani et al. [58]. Briefly, the inactivation was performed by heating samples for 60 min at 56 °C and by adding freezing ethanol (Honeywell, ≥99.8%, Offenbach, Germany), 9:1 ethanol/sample (*v*/*v*).

After that, samples were centrifuged at 13,000 rcf at 4 °C for 15 min, then, the supernatant was discarded while the pellet was left to air-dry and resuspended in 50 mM ammonium bicarbonate (NH_4_HCO_3_, Sigma-Aldrich, ≥99.0%, Darmstadt, Germany) buffer solution.

To enhance enzymatic digestion, RapiGest^TM^ SF Surfactant (Waters Corporation, Milford, MA, USA) was added to the final concentration of 0.1%. After protein quantification with a Nanodrop spectrophotometer (NanoDrop OneC, Thermo Scientific, Wilmington, DE, USA), N-Glycosidase F (Roche Diagnostics, Mannehim, Germany) was added to the samples (10 U/100 µL of plasma) and then incubated overnight at 37 °C.

### 4.3. Protein Digestion

Samples were treated with DL-Dithiothreitol (DTT) (Sigma-Aldrich, St. Louis, MO, USA, ≥99.5%) at the final concentration of 40 mM and incubated for 45 min at 56 ℃. Then, iodoacetamide (IAA) (Sigma-Aldrich, St. Louis, MO, USA) at the final concentration of 30 mM was added, and the samples were left at room temperature for 30 min for the carbamidomethylation reaction.

The proteins were enzymatically digested by adding 8 μg/100 µL of trypsin (trypsin from the porcine pancreas, Sigma-Aldrich, St. Louis, MO, USA), then, samples were incubated overnight at 37 °C. The enzymatic reaction was stopped by adding trifluoroacetic acid (TFA) (Honeywell, Seelze, Germany) to a final concentration of 0.5% and eventually formic acid (FA) (LiChropur®, Merck KGaA, Darmstadt, Germany) to reach an acidic pH (<2) and RapiGest^TM^ SF Surfactant was removed by centrifuging samples at 13000 rpm for 10 min. The supernatant containing the peptides was collected, then, the volume was reduced using a vacuum centrifugal evaporator (Hetovac, Savant); dried peptides were resuspended in 55 µL of loading pump phase A (H2O:ACN:TFA 98:2:0.1). 

Peptide content was quantified by using a Nanodrop spectrophotometer (NanoDrop OneC, Thermo Scientific, Wilmington, DE, USA) and samples were desalted using Ziptip™ µ-C18 Pipette Tips (Merck Millipore Ltd., Sigma-Aldrich, St. Louis, MO, USA). The peptides were eluted with a solution of 80% acetonitrile and 0.1% formic acid (FA), dried using a vacuum centrifugal evaporator (Hetovac, Savant) and resuspended in 50 µL of loading pump phase A.

### 4.4. Mass Spectrometry Analysis

For each sample, 2 µg of tryptic peptides were injected into a Dionex UltiMate 3000 rapid separation (RS) LC nanosystem (Thermo Scientific, Sunnyvale, CA, USA) coupled with an Impact HD^TM^ UHR-qToF system (Bruker Daltonics, Bremen, Germany). The samples were loaded into a µ-precolumn (Thermo Scientific, Acclaim PepMap 100, 100 µm × 2 cm, nanoViper, C18, 3 µm) for a further desalting and concentration step; then, the peptides were separated in an analytical 50 cm nanocolumn (Thermo Scientific, Acclaim PepMap RSLC, 75 µm × 50 cm, nanoViper, C18, 2 µm) with a multistep 240 min gradient ranging from 4% to 98% of nanopump phase B (H2O:ACN:FA 20:80:0.08) at a flow rate of 300 nL/min and temperature of the column in the oven of 40 ℃. The eluted peptides were ionised using a nanoBoosterCaptiveSpray™ (Bruker Daltonics) source using heated nitrogen dry gas (T = 150 ℃; 3 L/min) enriched with acetonitrile (ACN) (Honeywell, ≥99.9%, Offenbach, Germany). The mass spectrometer was operated in DDA (Data Dependent Acquisition Mode), with automatic switching between full-scan MS and MS/MS acquisition, as already reported [58]. N2 was used as a gas for CID (collision-induced dissociation) fragmentation. The software automatically selected the number of precursor ions in order to fit into a fixed cycle time; the time between two subsequent MS acquisitions was 5 s. The charge of precursor ions ranged between +2 and +5, and precursor peaks above 1575 counts, in the 300–1221 and 1225–2200 *m*/*z* windows, were selected. IDAS (Intensity Dependent Acquisition Speed) and RT2 (RealTime Re-Think) functionalities were applied. In order to achieve an improvement of mass accuracy, the mass spectrometer was calibrated using a mix of ten standards with a known mass (MMI-L Low Concentration Tuning Mix, Agilent Technologies, Santa Clara, CA, USA) before the sample run sequence. In addition, a specific lock mass (1221.9906 *m*/*z*) and a calibration segment (at the first 15 min of the gradient) of 10 mM sodium formate (1% NaOH 1 M and 0.1% FA) cluster solution was used.

### 4.5. Data Processing

Raw data were elaborated by using Compass DataAnalysis^TM^, v.4.1 (Bruker Daltonics, Hamburg, Germany). The resulting mass lists were processed using Peaks Studio X-Plus (Bioinformatics Solutions Inc., Waterloo, ON, USA); a SARS-CoV and SARS-CoV-2 protein database (UniProt, pre-release dataset, was downloaded from UniProt (ftp://ftp.uniprot.org/pub/databases/uniprot/pre_release/, accessed on 6 October 2021)), combined with a human database (SwissProt, released March 2020; 562,755 sequences; 202,599,198 residues) and integrated to the search engine was used. The parameters were set as follows: trypsin as the enzyme, carbamidomethyl as the fixed modifications, deamidated-only N and oxidation (M) as the variable modifications, 20 ppm as the precursor mass tolerances and 0.05 Da for the product ions. A false discovery rate (FDR) ≤ 1% was applied to all the analyses and the proteins were considered identified only if they had at least one unique significant peptide (FDR ≤ 1%; *p*-value ≤ 0.05), while in the quantification, only proteins with at least two unique significant peptides were considered. Functional annotations were performed using STRING (https://string-db.org/, version 11.5, accessed on 16 November 2022) [59] and g:Profiler (https://biit.cs.ut.ee/gprofiler/gost, 30 December 2022) [17].

### 4.6. Statistical Analysis on Quantified Proteins

The area of the top three peptides of each quantifiable protein was used to perform a statistical analysis adjusted for gender and age. Some plasma proteins (keratin proteins, albumin and histone proteins) that are highly abundant in blood or possible contaminants, or with no clinical interest in our study, were excluded from the analysis.

Patients’ demographic characteristics (i.e., Age and Sex) are reported as median (I-III quartile) and frequencies, both overall and stratified by the two groups. Non-parametric Wilcoxon Rank-Sum test and Exact Fisher test were performed on Age and Sex, respectively, in order to detect differences within “Mild” and “Severe”. 

In order to see which proteins were differentially expressed between the two groups, we built a volcano plot performing the non parametric Wilcoxon Rank-Sum test for clustered data and considering a *p*-value lower than 0.05 and a fold change of 1.5. 

Due to the high degree of biological variability, the higher amount of comparison with respect to the patient cohort, no further statistical correction was performed. For every considered protein feature the influence of sex and age was assessed in all the populations and a strict matching between Mild and Severe patients was considered stratifying the two groups by Age and Sex in order to adjust the influence of these confounding factors and improve the specificity of the disease signatures. 

To evaluate the similarities between the two groups, we built a heatmap with all of the proteins that were significantly different by using the area of the top three peptides of each protein after standardisation. 

Next, to select the most impactful proteins that discriminate between a mild and a severe score in blind and in order to find a threshold for the most discriminant proteins selected, we resorted to a classification tree which considered both all of the patients, as well as only the patients whose samples were collected in the first 21 days from the positive swab (patients with negative test excluded); the data were processed by the function rpart in R. In order to maximise classification accuracy, we controlled this aspect by changing the control parameters named *minsplit*, which define the minimum number of observations that must exist in a node in order for a split to be attempted. We define *minsplit* equal to 2.

All statistical analyses were performed using the open-source R software v.4.1.3 (R Foundation for Statistical Computing, Vienna, Austria).

## 5. Conclusions

An untargeted exploration of the plasma proteome changes associated with COVID-19 severity was conducted and the post-processing statistical elaboration returned a panel of 29 dysregulated proteins; 17 over-expressed in severe hospitalised patients in need of oxygen supply, and the remaining 12 in subjects with no symptoms or mild outcomes. The biological role and data from the literature support the molecular contextualization of these signatures and enrich the overview of the molecular signatures connected to the severity of this disease, from the expected increase of CRP in SEVEREs, to the modulation of specific proteins involved in humoral and cell-mediated immune response or in the hyper-coagulation process.

Moreover, by a supervised selection, the abundance of Fetuin-A combined with those related to Immunoglobulin lambda-2 chain C regions and Vitronectin emerged to be a key parameter able to distinguish severe patients from to mild ones in our cohort, with no mismatch, independently from the time of collection, the stage of the infection, gender and age.

The functional insight of the enriched pathways based on the deregulated proteins describes a more intriguing and entangled molecular picture in which the overexpression of differential proteins seems to map in pathways enriched only for severe patients (as classical antibody-related complement activation, the adaptive immune response by BCR signalling, innate immune response via activation and signalling of FCGR and FCERI and the scavenging of heme from plasma) or for both SEVEREs and MILDs (as regards the coagulation cascade and the regulation of the complement), but not only for mild patients. In addition, focusing on the SARS-CoV-2 signalling pathway, it has been noted that the proteins upregulated in SEVEREs refer to positively regulated virus-mediated pathways (such as SAA1/2, CRP, HP, LRG1), while proteins overexpressed in MILDs (GSN and HRG) are connected to processes that are expected to be negatively regulated in COVID-19, suggesting a potentially protective role of these molecules for serious outcomes. 

Taken together, these pieces of evidence likely suggest that the hyper-modulation of critical protective nodes in MILDs or of the triggers in SEVEREs, together with the absence of specific barrier points in SEVEREs, could drive the domino effect of biological processes leading to the worsening of clinical manifestations. 

## Figures and Tables

**Figure 1 ijms-24-03570-f001:**
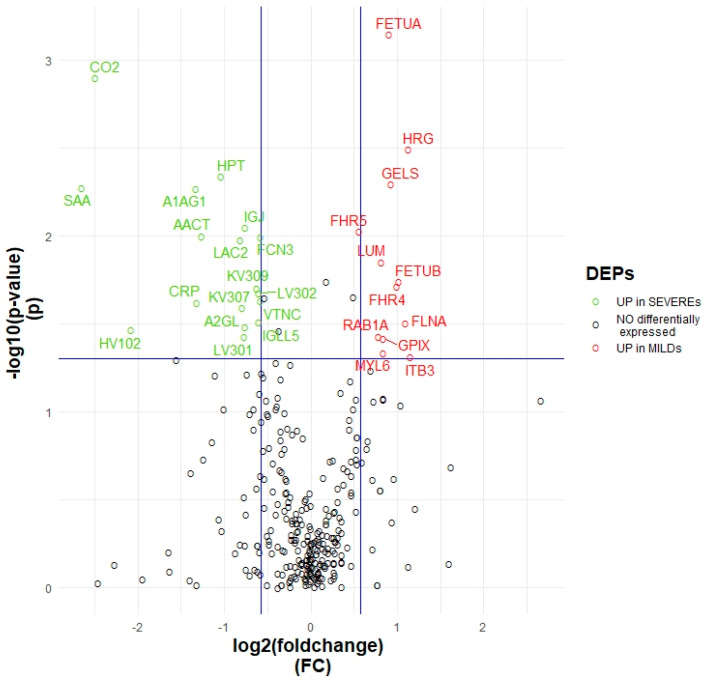
Volcano plot. Volcano plot showing which proteins are significantly deregulated between MILDs and SEVEREs considering a *p*-value (*p*) ≤ 0.05 and a fold change threshold of ±1.5. 17 proteins, in green, are upregulated in SEVEREs (Fold change ≤ −1.5); while 12 proteins, in red, are upregulated in MILDs (Fold change ≥ 1.5). Proteins are reported as UNIPROT_ID.

**Figure 2 ijms-24-03570-f002:**
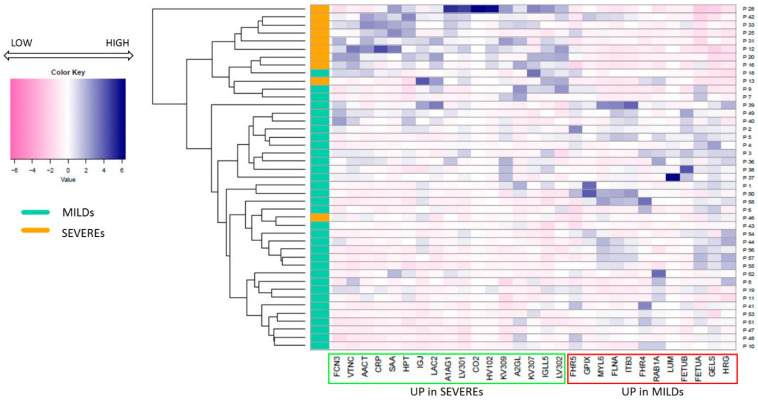
Heatmap-cluster analysis. The heatmap shows the different protein expression levels in correlation to each patient (reported as proteomic IDs). The green box highlights the proteins that are up-expressed in Severe patients, while proteins up-expressed in Mild patients are highlighted by a red box; it can be observed that the two groups of proteins differentially clusterise. Patients belonging to the MILDs group are shown in aquamarine, while SEVEREs are shown in orange; the colour scale from pink to blue indicates the level of expression of each protein, from a low expression to a high expression.

**Figure 3 ijms-24-03570-f003:**
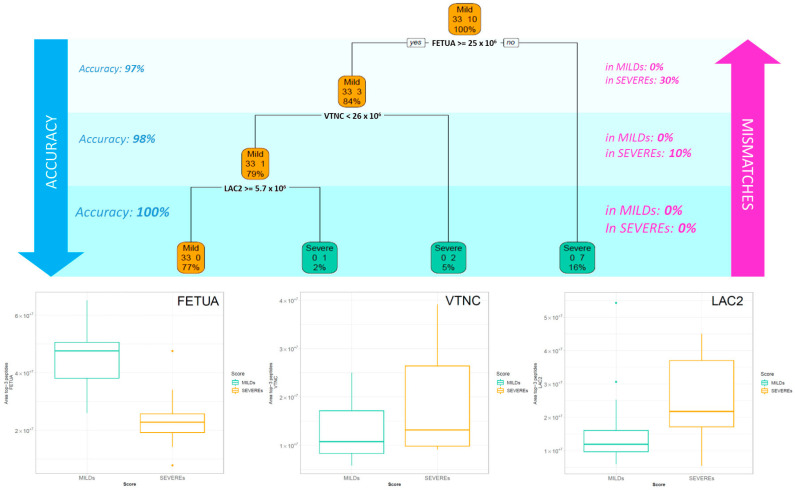
Decision tree of all 43 patients. The decision tree (DT) selects the most impactful proteins with the relative cut-off able to discriminate between the two classes. In each box are reported: the more frequent class indicated by the caption “Mild” or “Severe,” the number of cases belonging to the two classes and the percentage of patients in that box. The boxplot of the three selected proteins by the DT are reported for Mild and Severe patients.

**Figure 4 ijms-24-03570-f004:**
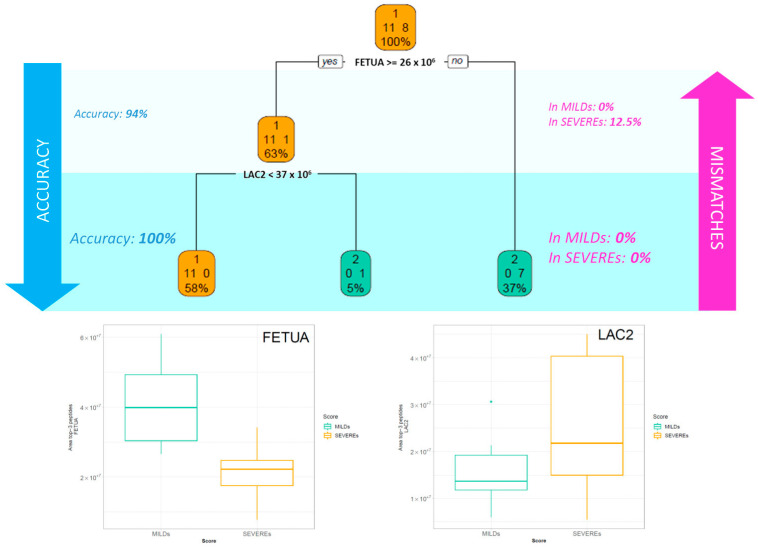
Decision tree of patients during the acute phase. The picture illustrates the supervised decision tree (DT) analysis related only to the patients whose samples were collected in the acute phase or at least within 21 days from the first positive test (patients with negative test excluded). In each box are reported: the more frequent class indicated by the caption “Mild” or “Severe”, the number of cases belonging to the two classes and the percentage of patients in that box. The boxplot of the three selected proteins by the DT are reported for Mild and Severe patients.

**Figure 5 ijms-24-03570-f005:**
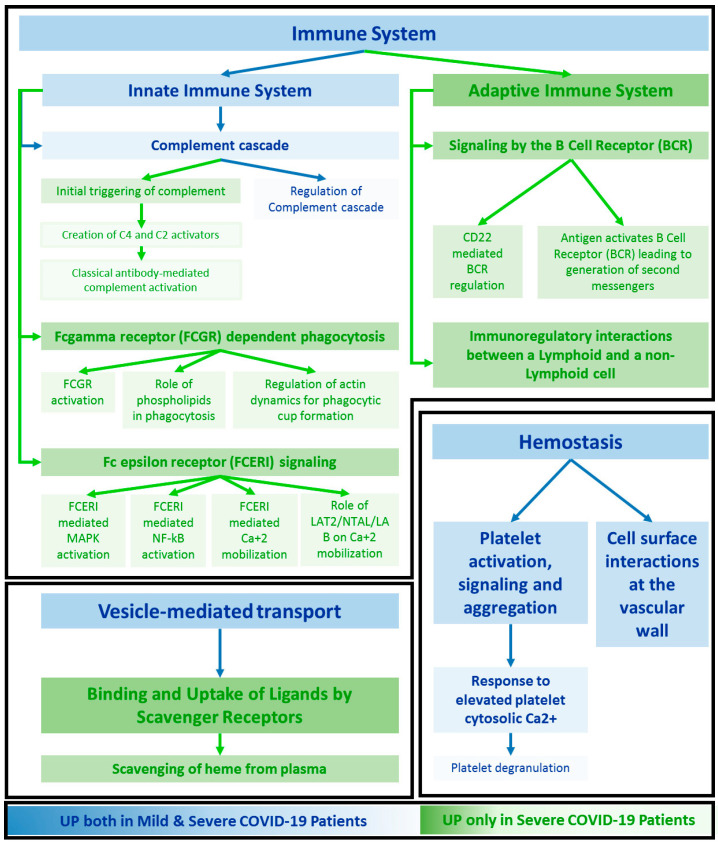
Network of the enriched pathways. Overview of the main significant pathways resulting from enrichment based on the 29 DEPs in Mild and Severe SARS-CoV-2 patients (g:Profiler [17]). The network is illustrated following the hierarchical structure of the Reactome database. Pathways which are enriched only in SEVEREs are in green, while those expressed in both MILDs and SEVEREs are in blue.

**Figure 6 ijms-24-03570-f006:**
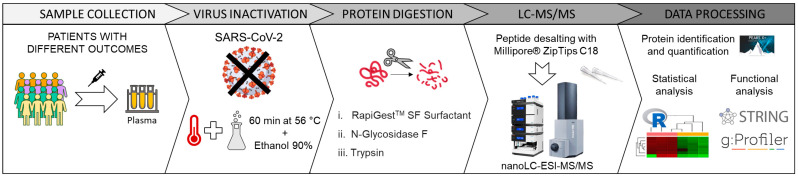
Experimental design. Brief illustration of the workflow, from the collection of the samples from COVID-19 patients with different outcomes to the statistical and functional analysis. After the collection, samples undergo an inactivation process of SARS-CoV-2 virus, then, proteins are deglycosylated and enzymatically digested, the peptide mixture is analysed by LC-MS/MS and data are finally processed by specific software.

**Table 1 ijms-24-03570-t001:** Cohort description—age and gender. This table summarises the demographic characteristics of the 43 patients selected for the analysis: Age and Sex overall and stratified by the WHO Score, *p*-value and the corresponding statistical tests are reported.

		Overall	Mild	Severe			
			WHO Score 1	WHO Score 4	WHO Score 5		
Participants (n)		43	33	5	5	*p*	Test
AGE (median [IQR])		56.00 [39.50, 63.00]	47.00 [35.00, 60.00]	65.50 [58.75, 67.75]		0.014	Wilcoxon
SEX (%)	Female	20 (46.5)	16 (48.5)	4 (40.0)		0.728	Exact Fisher
	Male	23 (53.5)	17 (51.5)	6 (60.0)			

## Data Availability

The data presented in the study are available upon request from the corresponding author.

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
