# Peer review of "Plasma Proteomic Variables Related to COVID-19 Severity: An Untargeted nLC-MS/MS Investigation"

_ijms, 2023, doi:10.3390/ijms24043570_

Round 1

Reviewer 1 Report

Within the submitted manuscript, Pagani et al. explored the plasma proteome of 43 COVID-19 patients stratified by sex, age and disease severity (mild/severe). They extract differentially-expressed proteins from patients' plasma proteome with the potential to act as disease severity markers, evaluate their biological role at various degrees in silico, and train a Decision tree using these proteins to discriminate between disease classes.
Apart from some technical adjustments concerning statistics, and minor editing of the introduction, the work is worth publication.

My suggestion is to ACCEPT THE WORK WITH MINOR REVISIONS

General remark: Can the authors provide the R script(s) as supplementary materials OR explicitly declare all the non-default parameters used when calling different functions?

Line-specific comments:
52-55:  Given the explicit treatment of sex/age strata the authors give in their analysis and the acknowledgement of age-related issues in disease severity, I think it would be worth further detailing the dependence of severity from the environment (
https://doi.org/10.1016/j.rmed.2021.106356), diet (http://dx.doi.org/10.1136/gutjnl-2021-325353) and genetic background (https://doi.org/10.1016/j.ejmg.2021.104227).
100: The term "node" may sound obscure to non-expert readers. I would substitute it with a more appropriate term or give some context.
131-132: This phrase belongs to methods and could confound non-expert readers. I suggest removing it.
159-171: I fail to see the point of the heatmap: is not the column (i.e. protein) clustering trivial since the list derives from the volcano plot? Regarding the row (i.e. patient) clustering, the authors should explicitly display the dendrogram: the way the figure is presented convey the idea that no row clustering has been produced. See also my observations about the figure itself.
577: can the authors give more details on the sex/age-based adjustment?
 582-584 can the authors give the effect size or the confidence interval? Are they adjusting p-values during their differential expression analysis?
298-300: Given the performance of the FETUA protein as a discriminant reported in the manuscript, can the authors elaborate on the advantages of using a DT versus a (more straightforward) threshold on the former?

Figure S1: The figure caption states the colour encodes " the level of expression of each protein, from a low expression to a high expression", yet the value range does not convince me. Do the authors intend the "fold change" as "expression level"?

Figure 1:  The title is self-referencing and contains a typo (vUlcano in place of vOlcano). I suggest removing it altogether.
Besides, the linear y-axis is also unconventional: may the author convert it in -log10(p)?

Author Response

Within the submitted manuscript, Pagani et al. explored the plasma proteome of 43 COVID-19 patients stratified by sex, age and disease severity (mild/severe). They extract differentially-expressed proteins from patients' plasma proteome with the potential to act as disease severity markers, evaluate their biological role at various degrees in silico, and train a Decision tree using these proteins to discriminate between disease classes.

Apart from some technical adjustments concerning statistics, and minor editing of the introduction, the work is worth publication.

My suggestion is to ACCEPT THE WORK WITH MINOR REVISIONS

General remark: Can the authors provide the R script(s) as supplementary materials OR explicitly declare all the non-default parameters used when calling different functions?

Thank you for the input. The following sentence has been add in the Materials and Methods section 4.6: “In order to maximise classification accuracy, we control this aspect by changing the control parameters minsplit which define the minimum number of observations that must exist in a node in order for a split to be attempted. We define  minsplit equal 2.” 

Line-specific comments:

52-55:  Given the explicit treatment of sex/age strata the authors give in their analysis and the acknowledgement of age-related issues in disease severity, I think it would be worth further detailing the dependence of severity from the environment (https://doi.org/10.1016/j.rmed.2021.106356), diet (http://dx.doi.org/10.1136/gutjnl-2021-325353) and genetic background (https://doi.org/10.1016/j.ejmg.2021.104227).

Thanks for highlighting an important aspect that we have inserted in the introduction, line 53-56. As pointed out, these factors (diet, environment, genetic background), differently from age and gender were not taken into account. Indeed, our aim was the exploration of specific protein profiles able to characterise COVID-19 patients with severe or mild symptoms. In this perspective, only possible confounding factors and not the predisposing or high-risk factors were considered. As mentioned in Discussion 3.5 (Limitations), unfortunately due to the not complete metadata collection, only the influence of age and gender were measured and excluded.

100: The term "node" may sound obscure to non-expert readers. I would substitute it with a more appropriate term or give some context.

Thanks for your suggestion, we substitute the term “node” with the more appropriate term: biological networks.

131-132: This phrase belongs to methods and could confound non-expert readers. I suggest removing it.

Thanks for your suggestion, we have removed that sentence and integrated it  in Materials and Methods section 4.6.

159-171: I fail to see the point of the heatmap: is not the column (i.e. protein) clustering trivial since the list derives from the volcano plot? Regarding the row (i.e. patient) clustering, the authors should explicitly display the dendrogram: the way the figure is presented convey the idea that no row clustering has been produced. See also my observations about the figure itself.

We thank the reviewer for noticing this error. We corrected the Figure according to the observation.

577: can the authors give more details on the sex/age-based adjustment?

Thanks for raising this topic. Accordingly, we have elucidated this aspect concerning the adjustment for sex and age. For every considered protein feature in the volcano plot the influence of sex and age was minimised using a strictly matched between Mild and Severe, stratifying for these confounding factors (age and sex). This point has been better specified in Materials and Methods section 4.6. 

 582-584 can the authors give the effect size or the confidence interval? Are they adjusting p-values during their differential expression analysis?

First, patients were grouped according to Age and Sex, then proteins were considered to have a different abundance (fold-change ratio > log2(1.5) and <log2(1.5)) when the p-value yielded by the Wilcoxon test was less than -log10(0.05). Due to the high degree of biological variability, the higher amount of comparison with respect to the patient's cohort, no further statistical correction was performed. We better specify this point in Materials and Methods section 4.6.

298-300: Given the performance of the FETUA protein as a discriminant reported in the manuscript, can the authors elaborate on the advantages of using a DT versus a (more straightforward) threshold on the former?

To investigate the influence of all the proteins, we used a DT in order to blindly select the most ones able to discriminate Mild vs Severe. Moreover, with this method we were able to find a threshold for the most discriminant proteins selected.

Figure S1: The figure caption states the colour encodes " the level of expression of each protein, from a low expression to a high expression", yet the value range does not convince me. Do the authors intend the "fold change" as "expression level"?

Thank you for your note. The heatmap has been built using the area of the top 3 peptides for each protein  after standardisation (we have added this information in Materials and Methods section 4.6). Therefore the expression level is not referred to the fold change, it is referred to the area of the top 3 peptides.

Figure 1:  The title is self-referencing and contains a typo (vUlcano in place of vOlcano). I suggest removing it altogether.

Besides, the linear y-axis is also unconventional: may the author convert it in -log10(p)?

Thank you for the input. We modified Figure 1 according to your suggestions. 

We have also changed the x-axis converting it in log2, as required by another reviewer. Moreover, we add  names of significant proteins inside the plot, in order to give a correct correspondence between protein name, p-value and fold change.

Reviewer 2 Report

This paper identified potential plasma marker proteins that could predict the severeness of COVID-19 infection in 43 patient samples via LC-MS/MS. Though similar study has been published previously, this paper identified overlapping hits while providing VTNC, IGL2 and FETUA as 3 novel hits. The overall logic is coherent, data is processed and presented well. However, some corrections are needed as follows. 

Major adjustments:

The discussion part is too lengthy and repeated. Section 3.1 and 3.2 should be more concise (or even deleted), because it was stated in previous sections already. 3.4 should be cut half or splitted into result part. 

Table 2 should be put in supplement, while Fig S1 should be in the manuscript.

The subset "infected" (19 patients) seems confusing. Please provide more information on when all these 43 plasma samples were collected in supplement (n days after first tested COVID positive, and which year/month they were collected). 

In Table 1, it is obvious to see that age is significantly correlated with the severeness of the disease. More evidence should be given so that the proteins identified in this research are irrelevant to age, or at least mentioned in discussion part 3.6

More information (about the biological functions, etc) should be provided in discussion part 3.3 about the identified hits. It is also worth discussing the changed antibody V genes identified in these severe patients. 

More information about how many proteins were identified (protein coverage) in MS should be mentioned or included in supplement.

Minor mistakes that needs to be corrected:

There are a lot of typos and spelling mistake and the authors should read carefully to correct. 

Fig.1: (1) in the title there is a typo "vulcano" should be "volcano"

 (2) The y and x axis of this plot should be displayed as -log P and log2 Fold change respectively.

(3) Fig.2 and 3:  Please write a short legend for the bar charts (FETUA, LAC2) to explain what y axis represents. 

(4) misspelling "deglycosylation" for "deglycosilation" in method part.

(5) line 151, "up-gulated" should be "up-regulated"

(6) In your method section (section 4.3), the IAA concentration is lower than DTT, which may lead to incomplete alkylation of Cys if excess DTT is not removed. Excess IAA should be quenched with DTT after alkylation.

(7)Table S5: "hospitalized" and "hospitalised" should be spelled identical; "oygenation" should be "oxygenation"

(8) Fig. 4: "Ca+2 mobilization" and "calcium mobilization" "Ca2+"should be in uniform format

(9) line 57, "small invasive"  should be "less invasive"

(10) line 229 "enrolment" should be "enrollment"

Author Response

Comments and Suggestions for Authors

This paper identified potential plasma marker proteins that could predict the severeness of COVID-19 infection in 43 patient samples via LC-MS/MS. Though similar study has been published previously, this paper identified overlapping hits while providing VTNC, IGL2 and FETUA as 3 novel hits. The overall logic is coherent, data is processed and presented well. However, some corrections are needed as follows. 

Major adjustments:

The discussion part is too lengthy and repeated. Section 3.1 and 3.2 should be more concise (or even deleted), because it was stated in previous sections already. 3.4 should be cut half or splitted into result part. 

According to the reviewer, the manuscript, particularly the discussion, was edited and submitted to a general restyle to reduce possible redundancies. Section 3.1 was removed and 3.2 was merged with section 3.3 avoiding repetitions. Section 3.4 was completely adjusted, splitting into different paragraphs and making the structure more schematic and in general clearer for the readers.

Table 2 should be put in supplement, while Fig S1 should be in the manuscript.

We agree with the reviewer and accordingly we moved the related Table 2 about the 29 protein differences resulted from the comparison of Mild vs Severe into supplemental material from results (Table 2 changed in Table S2) and possible inaccuracies were fixed. Conversely Figure S1 changed in Figure 2. 

The subset "infected" (19 patients) seems confusing. Please provide more information on when all these 43 plasma samples were collected in supplement (n days after first tested COVID positive, and which year/month they were collected). 

Thank you for your input. We have added information about the first positive swab and the time of sample collection for each patient in table S6. We have also clarified how we selected the subset “infected” at line 200-202.

In Table 1, it is obvious to see that age is significantly correlated with the severeness of the disease. More evidence should be given so that the proteins identified in this research are irrelevant to age, or at least mentioned in discussion part 3.6

We agree with the reviewer that this point was not adequately developed and can be a source of  misinterpretations. For every considered protein feature in the volcano plot the influence of sex and age was minimised using a strictly matched between Mild and Severe, stratifying for these confounding factors (age and sex). This point has been better specified in Materials and Methods section 4.6 and it has been also mentioned in Discussion section 3.5. 

More information (about the biological functions, etc) should be provided in discussion part 3.3 about the identified hits. It is also worth discussing the changed antibody V genes identified in these severe patients. 

Thanks for your suggestion, we have enriched the background of the three key proteins found with DT strategy, including a dissertation about the expression of the gene/transcript of this antibody related to COVID in blood in Discussion section 3.1.

More information about how many proteins were identified (protein coverage) in MS should be mentioned or included in supplement.

As suggested by the reviewer, we have replaced the table containing all the proteins identified and quantified (Table S1) with a new one containing some more information: for each protein we have also calculated the average protein coverage considering only mild patients and only severe patients, the number of identified peptides and the number of unique identified peptides were added. The new table also rectifies the number of proteins identified limiting to the subset identified only in our patients as mentioned in the text.

Minor mistakes that needs to be corrected:

There are a lot of typos and spelling mistake and the authors should read carefully to correct. 

Thank you for your observation. We modified the text according to your revision.

Fig.1: (1) in the title there is a typo "vulcano" should be "volcano"

 (2) The y and x axis of this plot should be displayed as -log P and log2 Fold change respectively.

Thank you for the input. We modified Figure 1 according to your suggestions. We delete the title with the typo, according to the request of another reviewer.

Moreover, we have changed the Figure adding the names of significant proteins inside the plot, in order to give a correct correspondence between proteins name, p-value and fold change.

(3) Fig.2 and 3:  Please write a short legend for the bar charts (FETUA, LAC2) to explain what y axis represents. 

Thank you for the observation, we have added a label on the y axis of the bar charts.

(4) misspelling "deglycosylation" for "deglycosilation" in method part.

We thank the reviewer for highlighting this misspelling, we corrected it in the method part.

(5) line 151, "up-gulated" should be "up-regulated"

Thanks,  we modified it.

(6) In your method section (section 4.3), the IAA concentration is lower than DTT, which may lead to incomplete alkylation of Cys if excess DTT is not removed. Excess IAA should be quenched with DTT after alkylation.

Thanks for noticing. We generally agree with you about the use of an equal or lower concentration of DTT respect than the one needed for IAA. However,  the concentration of reagents reported in the manuscript refers to an optimised procedure for COVID-19 plasma samples that we have previously implemented and published (Pagani, L.; Chinello, C.; Mahajneh, A.; Clerici, F.; Criscuolo, L.; Favalli, A.; Gruarin, P.; Grifantini, R.; Bandera, A.; Lombardi, A.; Ungaro, R.; Muscatello, A.; Blasi, F.; Gori, A.; Magni, F. Untargeted Mass Spectrometry Approach to Study SARS-CoV-2 Proteins in Human Plasma and Saliva Proteome. BioChem 2022, 2, 64-82. https://doi.org/10.3390/biochem2010005). Indeed, the excess of DTT was basically motivated by the need of helping the solubilization of protein aggregates following the heat treatment operated for the complete virus inactivation. Moreover, as shown in the reference, we verify that with this protocol the recovery and the identification performances were unchanged or improved. 

(7)Table S5: "hospitalized" and "hospitalised" should be spelled identical; "oygenation" should be "oxygenation"

(8) Fig. 4: "Ca+2 mobilization" and "calcium mobilization" "Ca2+"should be in uniform format

(9) line 57, "small invasive"  should be "less invasive"

(10) line 229 "enrolment" should be "enrollment"

Thanks for your support. We corrected all the typos indicated above.

Reviewer 3 Report

This work presents LC-MS/MS plasma proteomic analyses of COVID-19 in mild vs severe cases. As discriminator, fetuin-A, Ig lambda-2chain-C-region, Vitronectin are detected by the decision three.

If possible, can you please present the results summary in a table format including the relevant age groups and comparison with the other studies of interest.  In relation, what I am wondering in general is:

1-) Do you think the expression related discrepancies reflect deviations due to the course of infection or due to the present health states of those individuals, due to age, or so, which might also have led to a severe vs mild diseases, in turn?

 2-) Which one is more in the relevant studies, commonities or differences, and what are the possible reasons if differences and discrepancies are more?  I am not asking about the commonities and differences between your study and the others, but I am asking about the commonities and differences between all studies, including yours.

Finally, it was mentioned that keratin, albumin, and histones were redundant plasma proteins and considered as contaminants.  Can you please elaborate it more, especially the keratins and histones?  If we look at human proteins in plasma (https://www.proteinatlas.org/humanproteome/blood+protein/proteins+detected+in+ms), keratins are about 113 mg/L in sum while histones are less than half mg/L.

Author Response

Comments and Suggestions for Authors

This work presents LC-MS/MS plasma proteomic analyses of COVID-19 in mild vs severe cases. As discriminator, fetuin-A, Ig lambda-2chain-C-region, Vitronectin are detected by the decision three.

If possible, can you please present the results summary in a table format including the relevant age groups and comparison with the other studies of interest.  In relation, what I am wondering in general is:

To better clarify the result, we have revised the discussion in order to avoid possible redundancies and repetitions, to highlight the main outcome and to facilitate the readers schematizing the findings and including an extensive comparison with literature. Regarding the subdivision based on age relevant groups, we excluded a possible interference of the ‘age’ variable on our dataset by the statistical strategy described in Materials and Methods section 4.6.

1-) Do you think the expression related discrepancies reflect deviations due to the course of infection or due to the present health states of those individuals, due to age, or so, which might also have led to a severe vs mild diseases, in turn?

Thanks for noticing this aspect. We completely agree with the reviewer that age could affect the analysis and the alterations of protein expression acting as a confounding factor. Indeed, a significant difference in terms of age was calculated inside our comparison groups (Table 1). However, we operated a statistical approach able to exclude this effect. We detailed and clarified this point better in Materials and Method section 4.6. The influence of other factors such as the time of collection and consequently the time of the disease course was taken into account, re-calculating the DT also in a subset of the population whose plasma was collected only in the acute phase (within 21 days) and the key proteins as Fetuin-A was confirmed (see Results section 2.2). Unfortunately, as mentioned in Discussion (section 3.5-Limitations), due to the not complete metadata collection, not all the factors (such as comorbidities) can be considered for the present study. However, the partial heterogeneity of our cohort resulted functional to our aim which was more focused on exploring specific protein signatures of the severity than on studying the disease progression. 

 2-) Which one is more in the relevant studies, commonities or differences, and what are the possible reasons if differences and discrepancies are more?  I am not asking about the commonities and differences between your study and the others, but I am asking about the commonities and differences between all studies, including yours.

Thanks for your comment. We noticed that the results that we obtained were well aligned with what reported in literature, confirming the validity of the approach. All the significant signatures, highlighted both at protein and functional level, were extensively compared with studies on COVID-19 severity so far reported (see Discussion section 3.1-3.2-3.3). The discussion, as mentioned before, was re-thought and re-adjusted in order to schematize and improve the clearness of the outcome and the similarities with literature. 

Finally, it was mentioned that keratin, albumin, and histones were redundant plasma proteins and considered as contaminants.  Can you please elaborate it more, especially the keratins and histones?  If we look at human proteins in plasma (https://www.proteinatlas.org/humanproteome/blood+protein/proteins+detected+in+ms), keratins are about 113 mg/L in sum while histones are less than half mg/L.

Thank you for having pinpointed this aspect. We have explained more in detail the reason why we have removed these proteins from our analysis at line 590-592.

Round 2

Reviewer 1 Report

The paper can be accepted in the present form